# Blackbody radiation shift assessment for a lutetium ion clock

K.J. Arnold [1], R. Kaewuam[1], A. Roy[1], T.R. Tan[1,2] & M.D. Barrett[1,2]

The accuracy of state-of-the-art atomic clocks is derived from the insensitivity of narrow optical atomic resonances to environmental perturbations. Two such resonances in singly ionized lutetium have been identified with potentially lower sensitivities compared to other clock candidates. Here we report measurement of the most significant unknown atomic property of both transitions, the static differential scalar polarizability. From this, the fractional blackbody radiation shift for one of the transitions is found to be $-1.36(9) \times 10^{-18}$ at 300 K, the lowest of any established optical atomic clock. In consideration of leading systematic effects common to all ion clocks, both transitions compare favorably to the most accurate ion-based clocks reported to date. This work firmly establishes $Lu^{+}$ as a promising candidate for a future generation of more accurate optical atomic clocks.

[1] Centre for Quantum Technologies, 3 Science Drive 2, Singapore 117543, Singapore. [2] Department of Physics, National University of Singapore, 2 Science Drive 3, Singapore 117551, Singapore. Correspondence and requests for materials should be addressed to K.J.A. (email: cqtkja@nus.edu.sg) or to M.D.B. (email: phybmd@nus.edu.sg)

D evelopment of stable and accurate time standards has historically been an important driver of both fundamental science and applied technologies. The recent decade has seen phenomenal progress in atomic clocks based on optical transitions such that several systems now demonstrate frequency inaccuracies approaching $10^{-18}$[1–3], two orders of magnitude better than state-of-the-art cesium fountain clocks that currently define the SI second[4]. Redefinition of the second is already under consideration[5,6], but is unlikely until consensus on a best optical standard emerges. A significant technical hurdle for achieving inaccuracies below $10^{-18}$ outside of a cryogenic environment is the systematic uncertainty due to the blackbody radiation (BBR) shift[7].

For an optical transition with static differential scalar polarizability $\Delta\alpha_0 \equiv \alpha_0(e) - \alpha_0(g)$, where $e$ and $g$ refer to the excited and ground states respectively, the BBR shift, $\delta\nu_{bbr}$, in Hz, is given by[8]

$$\delta\nu_{bbr} = -\frac{1}{2h}\langle E^2(T_0)\rangle\left(\frac{T}{T_0}\right)^4 \Delta\alpha_0(1 + \eta(T)) \quad (1)$$

Here $\langle E^2(T_0)\rangle = (831.945 \text{ V m}^{-1})^2$ is the mean-squared electric field inside a blackbody at temperature $T_0 = 300$ K, and $\eta(T)$ is a temperature dependant correction which accounts for the frequency dependence of $\Delta\alpha_0(\nu)$ over the blackbody spectrum[8]. Polarizabilities are reported in atomic units throughout which can be converted to SI units via $\alpha/h$ (Hz m$^2$ V$^{-2}$) $= 2.48832 \times 10^{-8}\alpha$ (a.u.).

Singly ionized lutetium has two candidate clock transitions with favorable clock systematics[9,10], $^1S_0 \leftrightarrow {}^3D_1$ and $^1S_0 \leftrightarrow {}^3D_2$. Theoretical estimates of $\Delta\alpha_{0,1} \equiv \alpha_0(^3D_1) - \alpha_0(^1S_0) = 0.5(2.7)$ a.u. and $\Delta\alpha_{0,2} \equiv \alpha_0(^3D_2) - \alpha_0(^1S_0) = -0.9(2.9)$ a.u. have been given[11,12], albeit with indeterminate sign and magnitudes limited by the error estimates. In this work, experimental measurements of both $\Delta\alpha_{0,1}$ and $\Delta\alpha_{0,2}$ are made and the BBR shifts assessed. At 300 K, the fractional BBR shifts are $-1.36(9) \times 10^{-18}$ and $2.70(21) \times 10^{-17}$ for the $^1S_0 \leftrightarrow {}^3D_1$ and $^1S_0 \leftrightarrow {}^3D_2$ transitions, respectively. The former is the lowest among all optical clock candidates under active consideration[13]. Finally, the prospects for $^{176}$Lu$^+$ as a frequency standard are discussed by comparison of its atomic properties to other leading clock candidates.

## Results

### Measurement methodology.
While neutral-atom clocks have performed high accuracy ($2 \times 10^{-5}$) measurements of $\Delta\alpha_0$ using precise dc electric fields[14,15], this technique cannot be applied for ion clocks. Ion clocks have employed various methods to determine $\Delta\alpha_0$: inference from micromotion-induced stark shifts[16], cancellation of second-order Doppler and Stark micromotion shifts for cases where $\Delta\alpha_0 < 0$[17], and extrapolation from measurements of $\Delta\alpha_0(\nu)$ at near infrared (NIR) laser frequencies[2,18]. These approaches are not suitable for measurements in $^{176}$Lu$^+$ because, respectively: micromotion-induced Stark shifts would likely be too small to measure accurately, the signs of $\Delta\alpha_{0,1}$ and $\Delta\alpha_{0,2}$ are unknown, and extrapolation from measurements at NIR frequencies would be inconclusive given the magnitude of dynamic variation of the polarizabilties[11].

The approach taken here is to measure $\Delta\alpha_0(\nu_m)$ at the mid-infrared frequency of $\nu_m = c/(10.6 \text{ μm})$, which is near to the peak of the blackbody spectrum at room temperature. The availability of high power CO$_2$ laser sources ensures measurable ac Stark shifts even for small polarizabilities. In addition, a differential measurement of the clock frequency using an interleaved servo technique[19] requires only one clock and eliminates uncertainties due to common-mode perturbations. This approach enables unambiguous measurement of the sign and magnitude of the

polarizabilities with an accuracy limited by the determination of the CO$_2$ laser intensity at the ion.

For linearly polarized light of frequency $\nu$, the ac Stark shift of a state $|^3D_J, F, m_F\rangle$ is given by[20,21]

$$\delta\nu = -\frac{1}{2h}\langle E^2\rangle\Big(\alpha_{0,J}(\nu) + \frac{C_{F,m_F}}{2}\alpha_{2,J}(\nu)(3\cos^2\phi - 1)\Big) \quad (2)$$

where $C_{F,m_F}$ is a state dependant scale factor[21], $\alpha_{0,J}(\nu)$ and $\alpha_{2,J}(\nu)$ are, respectively, the dynamic scalar and tensor polarizabilities, $\langle E^2\rangle$ is the mean squared electric field averaged over one optical cycle, and $\phi$ is the angle between the polarization and quantization axes. The quantization axis is defined by an applied magnetic field of ~ 0.2 mT and different values of $\phi$ are obtained by rotating this field with respect to the CO$_2$ laser polarization. Light shifts as a function of beam position are used to characterize the beam profile and hence intensity for a given power. Light shifts at different values of $\phi$ for the transitions shown in Fig. 1b, are then used to determine differential scalar polarizabilities, $\Delta\alpha_{0,J}$, and tensor polarizabilities $\alpha_{2,J}$ for the two clock transitions.

### Experiment description.
The experimental setup consists of a single $^{176}$Lu$^+$ ion confined to a linear Paul trap with identical construction as in ref.[22]. The trap is operated with an rf drive frequency of $\Omega/2\pi = 20.8$ MHz. Detection, cooling, and state preparation are performed on the $^3D_1 \to {}^3P_0$ transition at 646 nm, with repump lasers at 350 and 622 nm to clear the $^1S_0$ and $^3D_2$ states, respectively (see Fig. 1a). The $^1S_0 \leftrightarrow {}^3D_1$ clock transition at 848 nm is a highly forbidden magnetic dipole (M1) transition with an estimated lifetime of 172 h[23]; and the $^1S_0 \leftrightarrow {}^3D_2$ at 804 nm is a spin-forbidden electric quadrupole (E2) transition with a measured lifetime of 17.3 s[11]. Both 848 and 804 nm clock lasers are frequency offset locked to the same reference cavity which has finesse of 400,000 and 30,000 at the respective wavelengths.

The 10.6 μm radiation is produced by a 10 W CO$_2$ laser. An acousto-optic modulator is used to control the optical power at the ion and as an optical switch, demonstrating better than 30 dB extinction. Two ZnSe vacuum viewports provide optical access to the ion. Displacement of the beam in the $yz$-plane is achieved with mirrors on motorized translation stages, as shown schematically in Fig. 2a and detailed in Methods. The laser is linearly polarized along the $z$-axis. The optical power is monitored with a thermal power meter at the exit viewport and after a dichroic mirror. The measured transmissions of the ZnSe

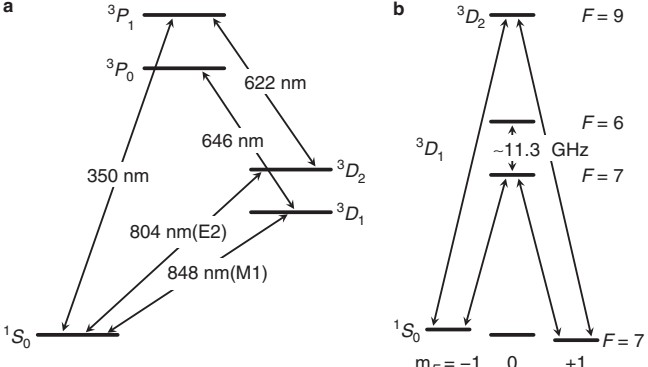

**Fig. 1** Energy level structure of Lu$^+$. **a** Low-lying energy levels showing the 804 and 848 nm clock transitions, 646 nm detection transition, and the 350 and 622 nm optical pumping transitions. **b** Hyperfine and magnetic substates for the clock transitions interrogated in this work, including the $F = 7$ to $F = 6$ microwave transition

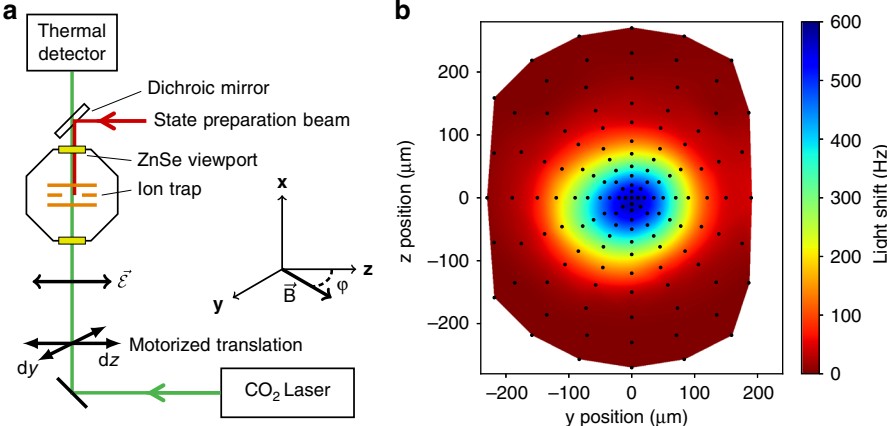

**Fig. 2** Schematic of $CO_2$ laser setup and measured beam profile. **a** The $CO_2$ laser is linearly polarized parallel to the z-axis. Motorized stages position the beam in the yz-plane (see Methods). The optical power is monitored after the vacuum chamber by a thermal detector. The magnetic field is rotated in the yz-plane to form an angle $\phi$ with respect to the fixed laser polarisation $\vec{\mathcal{E}} \parallel \mathbf{z}$. A $\pi$-polarized 646 nm state preparation laser counter-propagates to the $CO_2$ laser with the aid of a dichroic mirror. **b** Spatial profile of the laser at the ion's position measured via the light shift. Black points indicate measurement positions and the surface plot is a cubic spline interpolation of these measurements

viewport and dichroic mirror at 10.6 μm are 0.990 (5) and 0.986 (5), respectively.

The light shift induced on the $|^3D_1, 7, 0\rangle \leftrightarrow |^3D_1, 6, 0\rangle$ microwave transition has only a tensor contribution. This provides a useful diagnostic to determine the angle $\phi$ as the magnetic field is rotated: at $\phi = 90°$, the light shift is at an extremum, and, at $\phi \approx 54.7°$, the light shift vanishes. At $\phi \approx 90°$, the light shift measured as a function of beam position determines the spatial profile of the laser beam. From the measured profile shown in Fig. 2b, a peak intensity relative to the incident optical power of $I_{max}/P = 36.8\,(8)\,\text{mm}^{-2}$ can be inferred (see Methods). The interrogation sequence for the microwave measurements is the same as in ref.[22] with typical interrogation times of 10–50 ms depending on the desired Fourier limited resolution.

**Polarizability measurements**. To determine $\Delta\alpha_{0,J}(\nu_m)$, the magnetic field is rotated to $\phi \approx 54.7°$ where the tensor component of the light-shift vanishes. The 848 nm (804 nm) clock lasers are then stabilised to the average of the $|^1S_0, 7, \pm1\rangle \leftrightarrow |^3D_1, 7, 0\rangle$ ($|^3D_2, 9, 0\rangle$) transitions (see Fig. 1b). The light-shift is measured by interrogating alternately with and without the $CO_2$ laser and measuring the difference frequency between the two configurations. Typical interrogation times are 10–20 ms for the 848 nm transition and 1.5 ms for the 804 nm transition. Figure 3a shows the measured light shifts on the 848 nm clock transition (green points) as a function of $CO_2$ laser power together with interleaved measurements of the microwave transition (yellow points) for tracking the suppression of the tensor shift. The observed deviations of the microwave transition frequency are consistent with fluctuations of either 0.2 μT in the transverse magnetic field or ~ 1 mrad in the $CO_2$ laser polarization. The residual tensor shifts implied by the interleaved microwave measurements are removed from the optical measurements to generate the corrected data plotted in purple. A linear slope of $-9.60\,(13)\,\text{Hz W}^{-1}$ is deduced by a $\chi^2$ fit to the corrected data. Figure 3b shows the light shifts on the 804 nm clock transition at the same magnetic field orientation and has a fitted slope $201.3\,(2.2)\,\text{Hz W}^{-1}$. Owing to the larger light shifts involved, corrections due to residual tensor shifts were unnecessary.

The tensor polarizabilities are separately assessed by measuring the light shifts at $\phi = 90°$. Figure 3c shows the light shifts

measured on the $|^3D_1, 7, 0\rangle \leftrightarrow |^3D_1, 6, 0\rangle$ microwave transition with the fitted slope $530.4\,(3.4)\,\text{Hz W}^{-1}$. Figure 3d shows the light shifts on the $|^1S_0, 7, 0\rangle \leftrightarrow |^3D_2, 9, 0\rangle$ optical transition with the fitted slope $431.0\,(3.2)\,\text{Hz W}^{-1}$, from which the $^3D_2$ tensor polarizability can be determined by subtracting the previously determined scalar component.

From the data presented in Fig. 3, we evaluate the dynamic polarizabilities, in atomic units, to be

$$
\begin{aligned}
\Delta\alpha_{0,1}(\nu_m) &= 0.059(4) \\
\alpha_{2,1}(\nu_m) &= -4.40(34) \\
\Delta\alpha_{0,2}(\nu_m) &= -1.17(9) \\
\alpha_{2,2}(\nu_m) &= -4.53(40)
\end{aligned}
\tag{3}
$$

The largest sources of systematic uncertainty are the accuracy of the thermal power meter, specified at 5%, and an additional 6% effect which we attribute to etalon interference (see Methods). In evaluating $\Delta\alpha_{0,1}(\nu_m)$ we have included a 6.5% correction to account for the laser induced ac Zeeman shift (see Methods). We have not included consideration of the hyperfine mediated scalar polarizability for the $^3D_1$ levels[7,24]. From estimates omitting the hyperfine corrections to the wavefunctions, it is unlikely that this would be comparable to the measurement uncertainty. Nevertheless, it would be of interest to have a more accurate assessment given the size of the measured value.

**BBR shift analysis**. Because the measured differential scalar polarizability on the $^1S_0 \leftrightarrow {}^3D_1$ transition is very small, the extrapolation to dc and the dynamic BBR shift contribution must be carefully considered. Over the BBR spectrum near room temperature, the scalar polarizability is well described by a quadratic approximation

$$
\Delta\alpha(\nu) = \Delta\alpha_0(0) + (\Delta\alpha_0(\nu_m) - \Delta\alpha_0(0))\left(\frac{\nu}{\nu_m}\right)^2 \tag{4}
$$

For extrapolation to dc, we estimate the quadratic coefficient using the theoretical dipole transition matrix elements[11] and experimental energies[25]. Assuming 5% uncertainty in the theoretical matrix elements, we find $\Delta\alpha_{0,1}(0) = 0.018\,(6)$ (6)

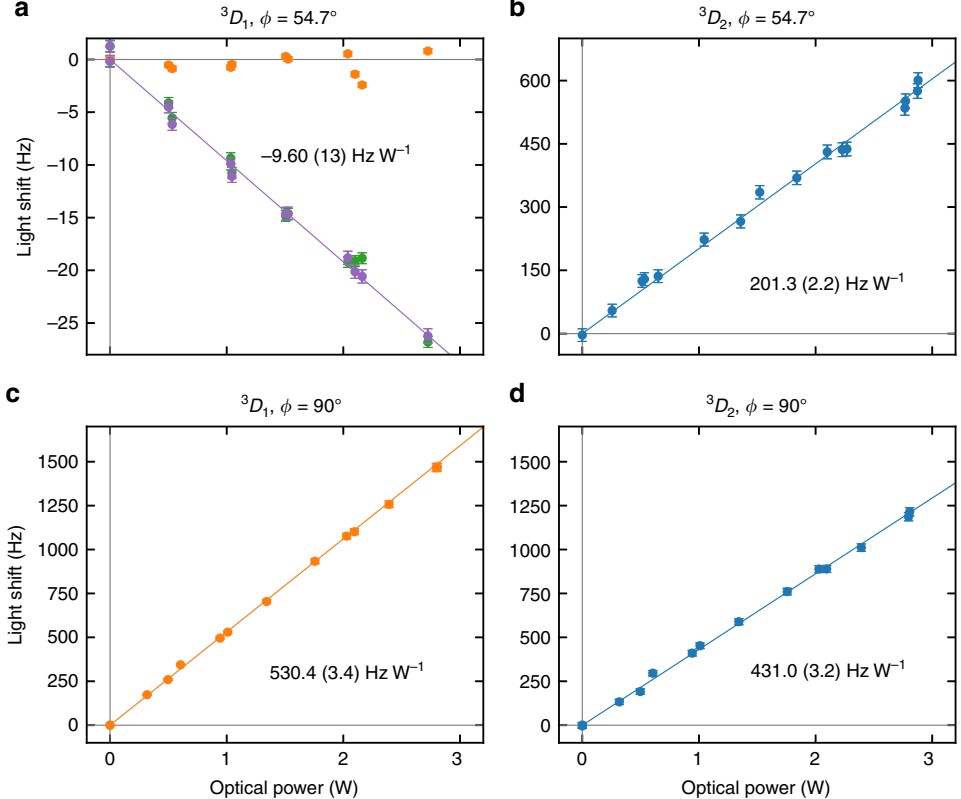

**Fig. 3** Measured lights shifts. For the $|^1S_0, F = 7, m_F = 0\rangle \leftrightarrow |^3D_1, 7, 0\rangle$ (green points), $|^1S_0, 7, 0\rangle \leftrightarrow |^3D_2, 9, 0\rangle$ (blue points) and $|^3D_1, 7, 0\rangle \leftrightarrow |^3D_1, 6, 0\rangle$ (orange points) transitions, the light shifts are measured as a function of incident power on the ion. Purple points are derived from the green data set corrected for residual tensor shift (see text). Error bars are the rms sum of contributions from the statistical servo error and 1.5% optical power instability. Solid lines are single parameter linear fits obtained by $\chi^2$ minimization

(a.u.). Evaluating the BBR shift by integrating the polarizability Eq. (4) over the BBR spectrum gives

$$
\delta\nu_{bbr} = -\frac{1}{2h}(831.945\,\mathrm{V\,m^{-1}})^2\left(\frac{T}{T_0}\right)^4 \times \left(\Delta\alpha_0(0)\right.
$$
$$
\left. +\beta(\Delta\alpha_0(\nu_m) - \Delta\alpha_0(0))\left(\frac{T}{T_0}\right)^2\right), \tag{5}
$$

where $\beta = \frac{40\pi^2}{21}\left(\frac{k_B T_0}{h\nu_m}\right)^2 \approx 0.918$. Direct comparison of this expression to Eq. (1) relates our measured dynamic polarizability to the usual dynamic correction factor, $\eta(T)$, up to quadratic terms. Because 10.6 μm is near to the center of the room temperature BBR spectrum, $\delta\nu_{bbr}$ is relatively insensitive to the dc value $\Delta\alpha_{0,1}(0)$. At ~ 313 K, $\delta\nu_{bbr}$ depends only on the measured value $\Delta\alpha_{0,1}(\nu_m)$. The BBR shift evaluated at 300 K is $-1.36(9) \times 10^{-18}$.

The insensitivity of the $^1S_0 \leftrightarrow {}^3D_1$ clock transition to temperature is illustrated in Fig. 4. Over the full range of 270–330 K the fractional uncertainty in the BBR shift remains below $1.0 \times 10^{-18}$. For more applicable laboratory conditions of $300 \pm 5$ K, indicated by the thin black lines, the fractional uncertainty is $2 \times 10^{-19}$.

At this level one might be concerned about the shift due to the magnetic field component of the BBR[7]. In this case the largest contribution is due to coupling to the $^3D_2$ state. The approximation used in ref.[7] for microwave transitions does not apply in this case and a numerical integration over the BBR spectrum must be used giving an estimated contribution of $3 \times 10^{-20}$ at room temperature. The hyperfine mediated scalar polarisability[7,24] will also contribute at the same order of magnitude with some cancelation due to hyperfine averaging[9] expected.

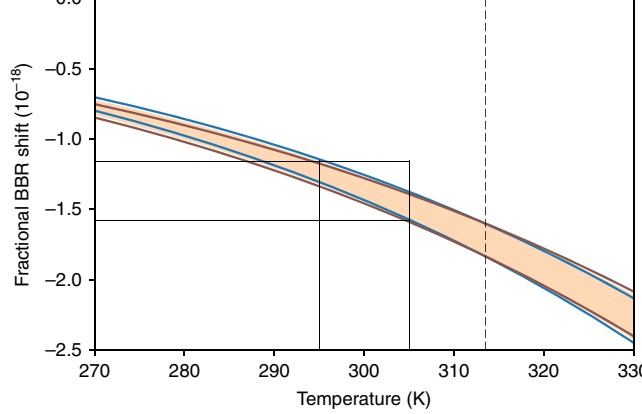

**Fig. 4** BBR shift on the $^1S_0 \leftrightarrow {}^3D_1$ transition. The shaded region is the total uncertainty from both the measurement, $\Delta\alpha_{0,1}(\nu_m)$, and extrapolation, $\Delta\alpha_{0,1}(0)$. The blue and red lines give the total uncertainty boundaries when $\Delta\alpha_{0,1}(0)$ is fixed at 0.012 (a.u.) and 0.024 (a.u.), respectively. Dashed line indicates the temperature at which $\delta\nu_{bbr}$ depends only on $\Delta\alpha_{0,1}(\nu_m)$. Solid lines indicate the operating condition of $300 \pm 5$ K and corresponding fractional frequency uncertainty of $2 \times 10^{-19}$

For the $^1S_0 \leftrightarrow {}^3D_2$ transition, the dynamic BBR shift is small compared to the scalar shift and we can approximate $\Delta\alpha_{0,2}(0) \approx \Delta\alpha_{0,2}(\nu_m)$. The corresponding fractional BBR shift is $2.70(21) \times 10^{-17}$ at 300 K.

For ion clocks, the value of $\Delta\alpha_0$ also has implications for micromotion-induced shifts. Micromotion driven by the rf-trapping field gives rise to two correlated clock shifts: an ac

Stark shift and a time dilation shift. The net micromotion shift, $\delta\nu_\mu$, to lowest order, is given by[17,26]:

$$\delta\nu_\mu = -\frac{\nu_0}{2}\left[\frac{\Delta\alpha_0}{h\nu_0} + \left(\frac{e}{mc}\right)^2\right]\left\langle E_\mu^2\right\rangle \tag{6}$$

where $\nu_0$ is the optical transition frequency in Hz, $e$ is the electron charge, $\Omega$ is the trap drive frequency, $m$ is the atomic mass, and $\left\langle E_\mu^2\right\rangle$ is the mean squared electric field at frequency $\Omega$. For clock transitions with $\Delta\alpha_0 < 0$, there exists a 'magic' drive frequency $\Omega_0 = \frac{e}{mc}\sqrt{-\frac{h\nu_0}{\Delta\alpha_0}}$ at which $\delta\nu_\mu$ vanishes. The suppression of micromotion shifts by operating at $\Omega_0$ has been applied in both $^{40}Ca^+$ and $^{88}Sr^+$ clocks[17,27] and is advantageous for multi-ion clock schemes[10,28]. An ideal clock candidate would have $\Delta\alpha_0 < 0$ which is small in magnitude to mitigate BBR shifts but sufficiently large to permit a practical value of $\Omega_0$. The $^1S_0 \leftrightarrow ^3D_2$ transition is in this ideal parameter regime with the lowest BBR shift among all such ion-clock candidates and a magic drive frequency of $\Omega_0/2\pi \approx 32.9$ (1.3) MHz.

## Discussion

With measurement of the differential scalar polarizabilities, all atomic properties of $^{176}Lu^+$ required to estimate clock systematics are known with sufficient accuracy to assess its future potential. The expected clock systematics for both transitions, after elimination of tensor shifts by hyperfine averaging[9], are summarized in Table 1. Achievable uncertainties in these systematic shifts are considered in reference to state-of-the-art experimental techniques. The $^1S_0 \leftrightarrow ^3D_1$ transition is uniquely insensitive to the BBR shift, leaving the residual micromotion-induced time dilation shift and ac-Stark shifts from the clock laser as leading systematics. Sensitivity to motional shifts favours heavier ions where, for example, evaluation of excess micromotion to the $10^{-19}$ level has been demonstrated for $^{172}Yb^+$[28]. Using hyper-Ramsey spectroscopy[29], suppression of the probe-induced ac Stark shifts by four orders of magnitude has been demonstrated in $Yb^+$[30], for which the shift is two order of magnitude larger than in $Lu^+$. The $^1S_0 \leftrightarrow ^3D_2$ transition has negligible probe-induced ac Stark shifts and net micromotion shifts can be heavily suppressed by operating near $\Omega_0$. With improved accuracy in the polarizability measurement[17] and a 1 K uncertainty in the temperature of the surroundings[2], the fractional BBR shift uncertainty could be reduced to the $10^{-19}$ level. For both $Lu^+$ transitions, second-order Doppler shifts due secular thermal motion are less than $10^{-19}$ for Doppler-limited cooling[10].

The overall clock systematics compare favorably to other candidates, including $^{171}Yb^+$[2] and $^{27}Al^+$[3], the two lowest uncertainty ion clocks reported at this time. The properties of the $^{171}Yb^+$ E3 clock transition offer no advantage over either $Lu^+$ transition in any category of Table 1: the BBR shift is ~3 times larger than for $Lu^+$ ($^3D_2$), micromotion considerations are comparable to $Lu^+$ ($^3D_1$), and the probe ac-Stark shift is two orders of magnitude larger than for $Lu^+$ ($^3D_1$) for the same probe time. $^{27}Al^+$ has a BBR shift ~6 times larger[3] than $Lu^+$ ($^3D_1$), requires an auxiliary ion for sympathetic cooling and state detection, and motion-induced shifts present a greater technical challenge due to the relatively light mass[31]. For experimental control comparable to that already demonstrated in the current generation of ion-based clock experiments, the systematics of $^{176}Lu^+$ suggest no significant hurdle for achieving evaluated uncertainties at the $10^{-18}$ level and beyond. Furthermore, its unique combination of atomic properties make $^{176}Lu^+$ a favoured candidate for multi-ion approaches[10,28] to advance the stability of ion-based atomic clocks.

## Methods

**Optomechanical setup for beam displacement.** The $CO_2$ laser propagates in free space and is directed onto the ion by two mirrors and a lens to focus the laser on the ion. Displacement in the z(y) - direction is achieved by translating both (one) mirrors and the lens. Motorised translation stages were characterized by detecting the position of a visible laser overlapped with the $CO_2$ laser using a CCD camera. The two axis of the motorized stages were found to be non-orthogonal by 4 mrad and this was subsequently corrected for in software. The rms positioning error measured by the camera was 0.7 (0.6) $\mu$m in the z (y) direction for a 2d profile scan comparable to the one shown in Fig. 2b.

**Analysis of the laser profile.** An approximate mode function is generated by cubic spline interpolation to the measurements shown in Fig. 2b. The peak intensity relative to the incident optical power is found by dividing the maximum light shift by the integral of this mode function over the interpolation region. The largest contributor to the uncertainty is the optical power instability of 1.5%. An elliptic $TEM_{00}$ gaussian model fit to the data indicates <0.5% of the light shift distribution is truncated by the finite interpolation region.

**Systematic uncertainties of the polarizabilities.** We observe discrete jumps in the measured light shift, monitored over the course of the day, at intervals of approximately one hour within a 6% range. These discreet jumps are not correlated with the optical power measured at the thermal detector. We attribute this predominately to frequency mode hops of the $CO_2$ laser which alters the effective transmission of the ZnSe viewport (dichroic mirror) between the thermal detector and ion by as much as 2% (2.7%) due to etalon effects. These frequency jumps occur on timescales longer than a typical data collection window. Because the $CO_2$ laser frequency is not controlled, we conservatively add the maximum range of variation observed in the light shift (6%) as an independent error to the power meter accuracy. Radiation reflected from the ZnSe window back to the ion contributes negligible (<0.1%) additional uncertainty in the intensity at the ion.

**Laser induced ac Zeeman shift.** Due to the small magnitude of $\Delta\alpha_{0,1}(\nu_m)$, shifts arising from the ac magnetic field of the $CO_2$ laser should be considered. The most significant contribution arises from the magnetic dipole coupling between the $^3D_1$ and $^3D_2$ fine structure states, which are separated by $\nu_0 \approx 19.15$ THz[22]. The quadratic Zeeman shift can be evaluated following the same treatment as for the dipole polarizability[21]. In the limit that the detuning is large relative to the hyperfine splitting, summation over all possible $F'$ results in a shift that can be broken down into scalar, vector, and tensor components. For linearly polarized light of frequency $\nu_m$ the ac Zeeman shift for state $|^3D_1, F, m_F\rangle$ is found to be

$$\delta\nu = -\left(\frac{1}{9}|\langle ^3D_2 \parallel \mathbf{M} \parallel ^3D_1\rangle|^2\right)\frac{\nu_0}{\nu_0^2 - \nu_m^2}\frac{\mu_B^2\langle B^2\rangle}{h^2}$$
$$\times \left(1 - \frac{C_{F,m_F}}{20}(3\cos^2\theta' - 1)\right) \tag{7}$$

where $|\langle ^3D_2\parallel\mathbf{M}\parallel ^3D_1\rangle|$ is the M1 reduced matrix element in Bohr magnetons, $\langle B^2\rangle$ is rms magnetic field, $C_{F,m_F}$ is the same state dependant coefficient as in Eq. (2), and $\theta'$ is the angle between the magnetic field and the quantization axis. Evaluated using the matrix element 2.055[11], we find the scalar shift on the $|^3D_1, 7, 0\rangle$ state is 0.627 Hz W$^{-1}$. This is subtracted from result shown in Fig. 3a to determine $\Delta\alpha_{0,1}(\nu_m)$. In this experiment the tensor component of electric field was nulled and consequently the tensor component of the orthogonal magnetic field was not cancelled. However, this magnetic tensor shift is sufficiently small that it does not impact significantly on the accuracy to which the tensor component due to the electric field was nulled. Contributions from other magnetic couplings, such as to the $^1D_2$ state, are not statistically significant.

**Table 1 Potential systematic shifts for $^{176}Lu^+$ clock transitions**

| Shift | $^1S_0 \leftrightarrow ^3D_1$ | $^1S_0 \leftrightarrow ^3D_2$ |
|---|---|---|
| Micromotion | † | 0 |
| Blackbody (300 K) | −1.36 | 27.0 |
| Probe ac Stark ($\tau_{\pi/2} = 50$ ms) | −460[a] | −0.15[b] |
| 2nd-order Doppler ($T = T_D$) | −0.08 | −0.05 |
| Quadratic Zeeman ($B = 10$ μT) | −1.36[b] | 0.48[b] |

Values are given as $10^{-18}$ fractional shifts of the respective transition frequencies. Hyperfine averaging[9] for both transitions, and operation at rf-trapping frequency $\Omega_0$ for $^3D_2$ are assumed. † dependant on characterization of excess micromotion. $T_D$ is the Doppler temperature for the 646 nm cooling transition
[a]ref.[23]
[b]ref.[11]

**Data availability**. All data presented and analyzed this study are available from the corresponding author upon reasonable request.

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

## Acknowledgements

We acknowledge J.S. Chen, S. Brewer and colleagues at the National Institute for Standards and Technology (NIST) for prompting us to evaluate the laser induced ac Zeeman shift. T. R. Tan acknowledges support from the Lee Kuan Yew postdoctoral fellowship. This research is supported by the National Research Foundation, Prime Ministers Office, Singapore and the Ministry of Education, Singapore under the Research Centres of Excellence programme. It is also supported by A*STAR SERC 2015 Public Sector Research Funding (PSF) Grant (SERC Project No: 1521200080).

## Author contributions

K.J.A., R.K., A.R., and T.R.T. performed the experiments. K.J.A. analysed the data and wrote the manuscript. M.D.B. conceived and supervised the project. All authors contributed to the construction of the experimental apparatus, discussed the results, and commented on the manuscript.

## Additional information

**Competing interests:** The authors declare no competing interests.

