## [Peer Review File · Nature Communications]

Reviewers' comments:

Reviewer #1 (Remarks to the Author):

This is a nicely written paper arguing advantages of Lu⁺ ion as the next-generation clock candidate. The experimental result seem to be sound and the arguments are well presented. The authors should address my questions/recommendations below.

At this level of E1 BBR suppression, one should start worrying about M1 contributions to the BBR shift. The authors should estimate M1 contribution to BBR. One of the issues is that Lu⁺ has hyperfine structure, so that M1 transitions to the neighboring hyperfine levels will contribute with small denominators, potentially enhancing the usually neglected M1 contribution. Order of magnitude estimate would suffice for the goals of this paper.

The authors also should add a discussion of how they would deal with quadrupolar shift, which should be a large contribution for the D_J levels - I presume they will use averaging technique over multiple transitions or use quantization axis averaging. Please spell it out as this is important for the overall clock performance estimate.

What is the expected stability of this single-ion clock?

Reviewer #2 (Remarks to the Author):

The manuscript describes measurements of the scalar and tensor polarizabilities for two optical transitions in Lu⁺ via the light shift of an infrared (10 micron) laser. The results are used to infer the light shift from blackbody radiation (BBR) around room temperature, an important parameter for the use of these Lu⁺ transitions as the reference of an optical clock. Theoretical estimates have been calculated earlier but did not reach the required precision, leaving the sign of the differential polarizabilities unknown. The experimental result shows that the BBR shift for the 1So - 3D1 transition is the lowest among the candidates for optical clocks that have been evaluated experimentally so far. The precise quantitative data on the polarizabilities make it possible to predict expected uncertainty contributions from the rf trapping field, another parameter of interest for the Lu⁺ trapped ion optical clock.

The experiment is a very well conceived and executed application of the state of the art in precision spectroscopy with trapped ions, applied to a novel system Lu⁺ that has been pioneered in the group of the authors. The analysis is sound and the paper is written very clearly. I recommend the manuscript for publication.

I have only a few suggestions for improvement and clarification of details:

page 2: The 1S0-3D1 clock transition is described as 'highly forbidden magnetic dipole (M1)'. It sounds a bit strange because M1 is a low multipole order. Maybe 'highly forbidden' could be explained as 'spin forbidden' or 'intercombination forbidden'.

The angle phi is predominately given in degrees (like 54.7, 90) but sometimes in radian (pi/2).

page 2: From the transmission of the ZnSe viewports one sees that about 2% of the IR laser power may be deposited there and could lead to heating. Since the thermal radiation emitted by the

viewports is not directed, the ion will see only a small fraction of it, but the spectrum of this radiation would be different from those of the laser and of the rest of the apparatus. Could it have an influence on the measurement?

page 2: It is mentioned that the microwave transition $F=6-F=7$ in the $3D1$ level 'has only a tensor contribution'. Wouldn't one also expect a small scalar hyperfine Stark effect like discussed in Itano et al., Phys. Rev. A 25, 1233 (1982)? It would not change the conclusions that follow this statement on page 2.

page 3: in the presentation of the data (eq. 3) it may be helpful to remind the reader that the numbers are given in atomic units. (Theory values on page 1 are written with the indication a.u.).

Reviewer #3 (Remarks to the Author):

The authors present a measurement of the room temperature blackbody radiation (BBR) shifts for two clock transitions in $^{176}\text{Lu}^{+}$. The results are generally interesting to the optical clock community because they demonstrate the feasibility of making a multi-ion optical clock with low sensitivity to the BBR environment while simultaneously minimizing the influence of ion micromotion. The technical convenience of the lasers involved, as well as the large ion mass, make $^{176}\text{Lu}^{+}$ an interesting candidate as a potentially competitive optical clock to $^{27}\text{Al}^{+}$ and $^{171}\text{Yb}^{+}$ or neutral atom lattice clocks based on Sr or Yb.

The authors present a measurement of the differential scalar polarizabilities for two transitions using a $10.6\ \mu\text{m}$ wavelength laser which is close in frequency to the peak of the BBR spectrum at room temperature. The evaluation of the beam profile and laser polarization are the dominant systematic uncertainties in the measurements.

Overall, the manuscript is clear and well-referenced and the results are new and of general interest. I recommend that the manuscript be accepted provided that the following questions/comments are addressed.

1. Have the authors considered the effect of the laser-induced quadratic Zeeman shift on the measured clock transition? The high-intensity laser used to induce an ac Stark shift on the clock transitions will also induce a quadratic Zeeman shift. In some cases, this shift can be a significant fraction ($\sim 10\%$) of the observed shift. If the observed shift is attributed entirely to the Stark effect, then the differential polarizability will be overestimated. This issue should be evaluated and added to the manuscript.
2. The B-field direction in Fig. 2a is unclear. Based on the figure and the text, it is now clear how the B-field orientation is being changed. The motorized stages are indicated, but it's not clear what is mounted on these stages. Is there a lens that is being translated? When the B-field orientation is changed how is this done? Is there a waveplate that is rotated, etc? Without this information it would be difficult for another group to reproduce these measurements.
3. Page 2, paragraph 3 "The $10.6\ \mu\text{m}$ radiation ..." What is the extinction ratio for the AOM used to shutter the laser? Has the influence of leakage light been considered? If so, an estimate should be included.

4. Page 4, "The suppression of micromotion ..." The claim that operation at a "magic" drive frequency "is the essential requirement for a multi-ion clock proposal" is not valid. This is a technically convenient feature of one of the transitions, but is not a requirement for multi-ion clock development. Given the level of progress that has been made in recent years in reducing micromotion-induced time-dilation uncertainty (i.e. ref. 29), I would advise the authors against making such claims. Instead, I would suggest that this feature be highlighted and not considered a requirement of multi-ion clocks.

5. TABLE I. The authors need to emphasize the fact that the systematic shifts reported here are estimates only and not strict evaluations. The magnitudes of the shifts reported here are given under certain assumptions that may or may not apply during actual clock operation. In addition, the fact that there are no associated uncertainties reported in the table is a problem. In particular, for the micromotion-insensitive transition, the BBR shift of 2.7×10^{-17} is the dominant systematic shift. However, similar systems operated at the "magic" drive frequency have achieved a micromotion-induced time-dilation shift uncertainty of 1.1×10^{-17} [27]. Taking the claimed room-temperature uncertainty of 2.1×10^{-18} in this BBR shift, and an estimated uncertainty of 1.1×10^{-17} for the micromotion time-dilation shift based on other work, one would infer a clock accuracy of approximately 1.1×10^{-17} . This level of accuracy would be comparable to, but not better than clocks in references [1,2,3]. Given the fact that the results could be interpreted in such a way, I advise the authors to include uncertainty estimates in TABLE I.

To the editor and referees,

An abstract and topical headings have been added to conform to Nature Communications style. Other changes are noted in the respective responses to referee comments.

We thank all referees for taking the time to carefully review our manuscript and providing helpful critical feedback.

We list the full referee comments verbatim, adding only labels to itemise our replies.

REFeree COMMENTS

Referee A

This is a nicely written paper arguing advantages of Lu+ ion as the next-generation clock candidate. The experimental result seem to be sound and the arguments are well presented. The authors should address my questions/recommendations below.

- **A1)** At this level of E1 BBR suppression, one should start worrying about M1 contributions to the BBR shift. The authors should estimate M1 contribution to BBR. One of the issues is that Lu+ has hyperfine structure, so that M1 transitions to the neighbouring hyperfine levels will contribute with small denominators, potentially enhancing the usually neglected M1 contribution. Order of magnitude estimate would suffice for the goals of this paper.
- **A2)** The authors also should add a discussion of how they would deal with quadrupolar shift, which should be a large contribution for the D_J levels - I presume they will use averaging technique over multiple transitions or use quantization axis averaging. Please spell it out as this is important for the overall clock performance estimate.
- **A3)** What is the expected stability of this single-ion clock?

Referee B

The manuscript describes measurements of the scalar and tensor polarizabilities for two optical transitions in Lu+ via the light shift of an infrared (10 micron) laser. The results are used to infer the light shift from blackbody radiation (BBR)

around room temperature, an important parameter for the use of these Lu+ transitions as the reference of an optical clock. Theoretical estimates have been calculated earlier but did not reach the required precision, leaving the sign of the differential polarizabilities unknown. The experimental result shows that the BBR shift for the 1S_0 - 3D_1 transition is the lowest among the candidates for optical clocks that have been evaluated experimentally so far. The precise quantitative data on the polarizabilities make it possible to predict expected uncertainty contributions from the rf trapping field, another parameter of interest for the Lu+ trapped ion optical clock. The experiment is a very well conceived and executed application of the state of the art in precision spectroscopy with trapped ions, applied to a novel system Lu+ that has been pioneered in the group of the authors. The analysis is sound and the paper is written very clearly. I recommend the manuscript for publication.

I have only a few suggestions for improvement and clarification of details:

- **B1)** page 2: The 1S_0 - 3D_1 clock transition is described as 'highly forbidden magnetic dipole (M1)'. It sounds a bit strange because M1 is a low multipole order. Maybe 'highly forbidden' could be explained as 'spin forbidden' or 'intercombination forbidden'.
- **B2)** The angle phi is predominately given in degrees (like 54.7, 90) but sometimes in radian (pi/2).
- **B3)** page 2: From the transmission of the ZnSe viewports one sees that about 2% of the IR laser power may be deposited there and could lead to heating. Since the thermal radiation emitted by the viewports is not directed, the ion will see only a small fraction of it, but the spectrum of this radiation would be different from those of the laser and of the rest of the apparatus. Could it have an influence on the measurement?
- **B4)** page 2: It is mentioned that the microwave transition $F=6$ - $F=7$ in the $3D_1$ level 'has only a tensor contribution'. Wouldn't one also expect a small scalar hyperfine Stark effect like discussed in Itano et al., Phys. Rev. A 25, 1233 (1982)? It would not change the conclusions that follow this statement on page 2.
- **B5)** page 3: in the presentation of the data (eq. 3) it may be helpful to remind the reader that the numbers are given in atomic units. (Theory values on page 1 are written with the indication a.u.).

Referee C

The authors present a measurement of the room temperature blackbody radiation (BBR) shifts for two clock transitions in $^{176}\text{Lu}^+$. The results are generally interesting to the optical clock community because they demonstrate the feasibility of making a multi-ion optical clock with low sensitivity to the BBR environment while simultaneously minimizing the influence of ion micromotion. The technical convenience of the lasers involved, as well as the large ion mass, make $^{176}\text{Lu}^+$ an interesting candidate as a potentially competitive optical clock to $^{27}\text{Al}^+$ and $^{171}\text{Yb}^+$ or neutral atom lattice clocks based on Sr or Yb.

The authors present a measurement of the differential scalar polarizabilities for two transitions using a 10.6 μm wavelength laser which is close in frequency to the peak of the BBR spectrum at room temperature. The evaluation of the beam profile and laser polarization are the dominant systematic uncertainties in the measurements.

Overall, the manuscript is clear and well-referenced and the results are new and of general interest. I recommend that the manuscript be accepted provided that the following questions/comments are addressed.

- **C1)** Have the authors considered the effect of the laser-induced quadratic Zeeman shift on the measured clock transition? The high-intensity laser used to induce an ac Stark shift on the clock transitions will also induce a quadratic Zeeman shift. In some cases, this shift can be a significant fraction (10%) of the observed shift. If the observed shift is attributed entirely to the Stark effect, then the differential polarizability will be overestimated. This issue should be evaluated and added to the manuscript.
- **C2)** The B-field direction in Fig. 2a is unclear. Based on the figure and the text, it is now clear how the B-field orientation is being changed. The motorized stages are indicated, but it's not clear what is mounted on these stages. Is there a lens that is being translated? When the B-field orientation is changed how is this done? Is there a waveplate that is rotated, etc? Without this information it would be difficult for another group to reproduce these measurements.
- **C3)** Page 2, paragraph 3 "The 10.6 μm radiation ..." What is the extinction ratio for the AOM used to shutter the laser? Has the influence of leakage light been considered? If so, an estimate should be included.
- **C4)** Page 4, "The suppression of micromotion ..." The claim that operation at a "magic" drive frequency "is the essential requirement for a multi-ion clock proposal" is not valid. This is a technically convenient feature of one of the transitions, but is not a requirement for multi-ion clock development. Given the level of progress that has been made in recent years in reducing micromotion-induced time-dilation uncertainty (i.e. ref. 29), I would advise the authors against making such claims. Instead, I would suggest that this feature be highlighted and not considered a requirement of multi-ion clocks.
- **C5)** TABLE I. The authors need to emphasize the fact that the systematic shifts reported here are estimates only and not strict evaluations. The magnitudes of the shifts reported here are given under certain assumptions that may or may not apply during actual clock operation. In addition, the fact that there are no associated uncertainties reported in the table is a problem. In particular, for the micromotion-insensitive transition, the BBR shift of $2.7\text{e-}17$ is the dominant systematic shift. However, similar systems operated at the "magic" drive frequency have achieved a micromotion-induced time-dilation shift uncertainty of $1.1\text{e-}17$ [27]. Taking the claimed room-temperature uncertainty of $2.1\text{e-}18$ in this BBR shift, and an estimated uncertainty of $1.1\text{e-}17$ for the micromotion time-dilation shift based on other work, one would infer a clock accuracy of approximately $1.1\text{e-}17$. This level of accuracy would be comparable to, but not better than clocks in references [1,2,3]. Given the fact that the results could be interpreted in such a way, I advise the authors to include uncertainty estimates in TABLE I.

RESPONSES

to Referee A comments

- **A1)** The hyperfine structure contribution to the M1 BBR shift is not significant: from Itano's paper [PRA 25:2,1982] the fractional frequency shift of a microwave transition is order $1\text{e-}17$ and its effect on an optical transition is suppressed by the ratio of the microwave to optical frequency. The larger issue would be the M1 couplings to the other fine structure levels. This can be treated in the same way but numerical integration over the spectrum must be used as the fine-structure splitting is central to the BBR spectrum. The shift goes to zero at a temperature which is dependent on the fine structure frequency splitting because part of the BBR spectrum shifts the energy up and part shifts

it down. In the case of Lutetium, we are near to this cancellation point for any realistic temperature and the shift is estimate to be $3e-20$ for both transition. Note that in the two extreme limits (microwave vs optical) the shift is either quadratic or quartic in temperature. Here we are somewhere in between.

A comment has been added to the manuscript regarding this effect.

- **A2)** It is already mentioned and cited in both the main text and in Table I caption that hyperfine averaging (manuscript Ref 11) will be used to address tensor shifts, which is inclusive of quadrupole shifts.
- **A3)** A single-ion clock is expected to be projection noise limited and thus comparable to any other single ion clock. Projection noise limited stability will be set by interrogation time and hence the laser stability. However, we note that our research goals are to develop a multi-ion strategy.

to Referee B comments

- **B1)** We have used the same terminology with which earlier theory papers have referred to this transition. Highly forbidden refers to the fact that an M1 transition itself is forbidden for the transition in question (by principle quantum number) and can only occur through electron configuration mixing (with an optically separated level). Hyperfine mixing (as in Al+, Sr and Yb) can and does give a much smaller quadrupole (E2 coupling).
- **B2)** All angles have been changed to degrees throughout for consistency.
- **B3)** The absorption by the ZnSe is negligible (0.0005 per cm at $10.6 \mu\text{m}$ is a typical specification). The windows are AR coated but there is still $\sim 0.5\%$ reflection per surface, hence the 0.990(5) transmission given on page 2. This was independently confirmed by measuring the reflected power from a ZnSe window at a small angle with respect to normal. Note the 2% number in Methods is in the context of etalon interference in reflections from these two surfaces, an effect which was also observed in independent tests on a identical window outside vacuum. One might be (and we were) concerned about how the reflected light from the window might affect the measurement. Considering the divergence from the $\sim 132 \mu\text{m}$ waist and propagation distance to the window ($\sim 8 \text{ cm}$) and back, the effect of this reflected power on the intensity at the ion is negligible ($< 0.1\%$). Comment added in manuscript to this effect.

- **B4)** Hyperfine mediated polarizabilities have not been calculated for lutetium to our knowledge so we could only crudely estimate this effect. If we take calculated matrix elements and estimate the effect by accounting for the hyperfine energy splittings we would expect values $\lesssim 1e-3$ (a.u.) - only the difference matters as we could absorb the mean value into the definition of the scalar. As noted in [PRA 82:062513 2010] hyperfine corrections to the wavefunctions contribute to the same order. At the level of $1e-3$, the effect on nulling the tensor would be below our measurement precision in nulling the shift (Hz out of 500Hz). The error in the angle would be on the order of the ratio of the differential hyperfine scalar polarizability to the tensor polarizability. The influence on the subsequent measurement on the F=7 scalar, which would fold back in the tensor component, would then be on the order of the differential hyperfine scalar polarizability between the F=6 and F=7 —specifically $5/7 \sim 0.7$ of this difference. It is unlikely that this would be comparable to the measurement uncertainty. However, we have added a line in the text to alert the reader to this potential issue and hope that it encourages theory colleagues to more accurately assess the effect.

- **B5)** manuscript updated accordingly

to Referee C comments

- **C1)** This issue was brought to our attention in conversation with colleagues at NIST while the manuscript was in review. We have evaluated this effect and it does indeed contribute at the level of the measurement uncertainty for the $^3D_1 \Delta\alpha_0(\omega_m)$. The manuscript has been updated accordingly and an acknowledgement added.
- **C2)** The caption of Fig2a has been modified to make clear that the laser polarization is fixed and the B field angle is variable. This is already clear in the main text. Fig2a indicates movement schematically but we do not see a good way to accurately depict the physical beam path and resulting translation. Instead we have added further description of the mechanical stages and the characterisation of the beam displacement in a Methods section which we believe is sufficient information for another experimentalist to recreate an equivalent setup. The magnetic field is rotated by changing the bias magnetic fields applied in the y and z directions. The measurements are not sensitive to the exact magnitude of magnetic field since the laser is always served to the average of two zeeman components. All stark shifts are measured differentially and any

quadratic Zeeman shifts due the static field are common mode.

- **C3)** To the minimum resolvable power ($\sim 2\text{-}3\text{ mW}$) of the more sensitive of the high-power thermal detectors we have on hand (Gentec UP19K-50L-H5-D0), the AOM switched off was indistinguishable from mechanically blocking the laser. This bounds the AOM extinction to at least better than 1000:1 such that it would not affect our results. Any leakage light at other wavelengths is common mode and would not affect the measurement result. Comment on the aom extinction added to the manuscript.
- **C4)** The claim as stated was strictly that a negative polarizability was essential for the ref 11 implementation. However, we agree it is unnecessary to restrict consideration to this particular implementation here. Wording and references have been altered accordingly to include other multi-ion implementations.
- **C5)** We have reworded the text to emphasise that we are giving a comparison of atomic properties relevant to clock operation but will refrain from making unsubstantiated claims of clock uncertainty specifically to avoid the interpretation the reviewer refers to. We feel there is no indication that we are making claims of current clock accuracy and it would be entirely inappropriate to do so. In previous work we have refrained from making comparisons to other clocks precisely because some properties cannot be accurately calculated, specifically the DC polarizability and the AC stark shift. With these now measured, the properties can now be faithfully stated. Comparison with the state of the art, and the consequent prospects for Lu^+ is completely reasonable. It is simply a matter of fact that an atom with a superior insensitivity to its environment will ultimately achieve better performance or, at the very least, be technically easier to implement assuming equal technical competency in the laboratory. We are thus making no claims about what we will achieve, but simply pointing out what those using Yb^+ , Al^+ , Sr^+ etc. might achieve if they turned their expertise toward Lu^+ .

With regards to the referee's specific micromotion example, it makes little sense to take the worst assessment of micromotion from the referenced literature [27] as a benchmark for potential performance in a Lu^+ clock. We note in [Ref27 : PRL 116, 013001 (2016)] the 'magic' frequency is determined from a theory value of the polarisability. Together with the lighter mass of Ca^+ , the effect is a $1\text{E-}17$ uncertainty. We have replaced this citation with [Appl. Phys. B (2017) 123:112] a more recent Ca^+ result which reports a factor 63 suppression

of micromotion shifts to the level of $1\text{E-}18$ ⁽¹⁾. A more reasonable point of comparison is [Sr^+ , Ref 18] which has already demonstrated 0.5% accuracy in polarisability measurement resulting in 200-fold reduction in micro-motion shifts at the 'magic rf' frequency. For Lu^+ (3D2), the accuracy of the polarisability reported in this work directly translates into a ~ 10 suppression factor for the micromotion shifts. Even at our current levels of micromotion compensation, for which we have not yet put in serious effort, we would conservatively estimate $\sim 1\text{e-}18$ in clock uncertainty (3D2) operating at the magic rf due o this effect. Ultimately the polarisability on the 3D2 will be more accurately assessed using the same technique as in [Sr^+ , Ref 18] and we anticipate 1% accuracy giving a 100-fold suppression of micromotion shifts. Measurements of this type are already planned for Lu.

For the other quantities in Table I:

Micromotion (3D1): While beyond the scope of the manuscript, we note that Lutetium has a tertiary clock transition at 577-nm (1S0-1D2) for which to probe micromotion effects with at least two orders of magnitude better (lifetime limited) interrogation time than the E2 transition in Yb^+ or the 1S0-3P1 used in Al^+ . The cooling transition is similarly an order of magnitude narrower leading to lower $\langle n \rangle$. Consequently there should be no limitation to achieving similar levels of micromotion compensation including thermal limitations of the sideband technique as discussed in [Ref 31] —again assuming equal laboratory skills. For the purpose of this manuscript, simply noting that realised uncertainty will be limited by experimental micromotion assessment while pointing out the benefit of a large mass and referencing the state of art [e.g. Ref 31], which what is already done the manuscript, seems appropriate.

Zeeman shifts: The values given in the table are taken from theoretical estimates. As these values have little dependence on the spatial wavefunctions, they are typically very accurate (a few %). We agree that they need to be measured, but they are not going to be substantially different from the values given insofar as comparing the sensitivity to different clock candidates is concerned.

AC stark shift: This is two orders of magnitude

¹ The assessment in this paper must be using a more accurate measured value of the polarisability but we are unable to find such a polarizability measurement for Ca^+ reported anywhere in the literature and [Appl. Phys. B (2017) 123:112] gives no reference. So their claimed suppression factor, while reasonable, appears to us to be unsubstantiated.

smaller than in Yb^+ , in which claims of $1\text{-}10^{-18}$ uncertainty in this shift are made. Achieving the same or better requires much less stringent operating

conditions for lutetium—in particular $\text{Lu}(3\text{D}1)$ will not be starkly shifted by more than the Fourier limited linewidth under reasonable interrogation scenarios.

REVIEWERS' COMMENTS:

Reviewer #1 (Remarks to the Author):

The response to my questions is adequate. I am recommending the acceptance.

Reviewer #2 (Remarks to the Author):

The authors have addressed the referee's comments very carefully and completely. This is a very well written paper reporting significant progress and insight. I recommend it for publication.

A final question about style: In the abstract we now find a statement '... lowest sensitivity to room-temperature blackbody radiation of any known clock'. This is not clear to me. What is a known clock? I may claim to 'know' 'clocks' that are likely to possess lower polarizabilities, for example in highly charged ions and in nuclear gamma transitions. The scientist reading this abstract in the future would probably be more interested in numbers than in such comparisons.

Reviewer #3 (Remarks to the Author):

The authors provide a revised version of the manuscript that has addressed all of my previous questions/comments. Therefore, I recommend that the manuscript be accepted.

To the editor and referees,

We thank referees for taking the time to review our manuscript. Referee comments are listed verbatim with our responses where required.

REFEREE COMMENTS

Referee 1

The response to my questions is adequate. I am recommending the acceptance.

Referee 2

The authors have addressed the referee's comments very carefully and completely. This is a very well written paper reporting significant progress and insight. I recommend it for publication.

A final question about style: In the abstract we

now find a statement '... lowest sensitivity to room-temperature blackbody radiation of any known clock'. This is not clear to me. What is a known clock? I may claim to 'know' 'clocks' that are likely to possess lower polarizabilities, for example in highly charged ions and in nuclear gamma transitions. The scientist reading this abstract in the future would probably be more interested in numbers than in such comparisons.

Referee 3

The authors provide a revised version of the manuscript that has addressed all of my previous questions/comments. Therefore, I recommend that the manuscript be accepted.

AUTHOR RESPONSE TO REFEREE 2

We agree that particular wording was not precise and so has been amended accordingly.